# Performance of High Efficiency Avalanche Poly-SiGe Devices for Photo-Sensing Applications

**DOI:** 10.3390/s22031243

**Published:** 2022-02-07

**Authors:** Yuang-Tung Cheng, Tsung-Lin Lu, Shang-Husuan Wang, Jyh-Jier Ho, Chung-Cheng Chang, Chau-Chang Chou, Jiashow Ho

**Affiliations:** 1Department of Electrical Engineering, National Taiwan Ocean University, No.2, Pei-Ning Rd., Keelung 202, Taiwan; yt5868@gmail.com (Y.-T.C.); zonglinlu1997@gmail.com (T.-L.L.); ricesimonwong@gmail.com (S.-H.W.); ccchang@mail.ntou.edu.tw (C.-C.C.); 2Department of Mechanical & Mechatronic Engineering, National Taiwan Ocean University, No.2, Pei-Ning Rd., Keelung 202, Taiwan; cchou@mail.ntou.edu.tw; 3Department of Electrical Engineering, University of California, Los Angeles, CA 90095, USA; jth72507@gmail.com

**Keywords:** poly-silicon germanium (poly-SiGe), low pressure chemical vapor deposition (LPCVD) system, responsivity, quantum efficiency, avalanche multiplication factor

## Abstract

This paper explores poly-silicon-germanium (poly-SiGe) avalanche photo-sensors (APSs) involving a device of heterojunction structures. A low pressure chemical vapor deposition (LPCVD) technique was used to deposit epitaxial poly-SiGe thin films. The thin films were subjected to annealing after the deposition. Our research shows that the most optimal thin films can be obtained at 800 °C for 30 min annealing in the hydrogen atmosphere. Under a 3-μW/cm^2^ incident light (with a wavelength of 550 nm) and up to 27-V biased voltage, the APS with a n^+^-n-p-p^+^ alloy/SiO_2_/Si-substrate structure using the better annealed poly-SiGe film process showed improved performance by nearly 70%, 96% in responsivity, and 85% in quantum efficiency, when compared to the non-annealed APS. The optimal avalanche multiplication factor curve of the APS developed under the exponent of *n* = 3 condition can be improved with an increase in uniformity corresponding to the APS-junction voltage. This finding is promising and can be adopted in future photo-sensing and optical communication applications.

## 1. Introduction

Traditionally, optical devices utilizing photomultiplier tubes (PMT) have been the mainstream [1,2,3,4], but with the development of applications, such as optical communication [1], 3-D image sensing [2], astronomical and biological detection [3,4], the demand for a high-performing, robust optical sensor has become more prevalent. The conventional PMT optical devices are oftentimes bulky, fragile, and sensitive to magnetic fields, thus requiring very high operation voltages and high-power consumption. In recent years, these disadvantages of PMT have put a limit on detection rate, high time resolution, and high spatial density [5].

A promising alternative to the PMT is the avalanche photosensor (APS), which has a simple structure and is compatible with the CMOS standard manufacturing process. It can be directly integrated in an IC chipset, which greatly reduces its size. Furthermore, the materials used in an APS have high-sensitivity, high-resolution, fast response-speed, low power-consumption, and relatively cheap manufacturing [6,7,8]. Presently, the silicon-based (Si-based) APS is one of the most popular photo-sensing technologies. This development technique can limit the sensitivity, response-speed, and device geometry of the end-product. By using a nano-opto-electro-mechanical system (NOEMS) fabrication process for integrated circuits (ICs), the APS allows the combination of light and electronic processing circuits (such as amplifier and filter circuits) to improve sensitivity [9,10]. However, while speed can be increased by reducing the time it takes to pass through the thin photo-sensing layer, it is more difficult to collect additional light in a single-chip Si-based system. This leads to a compromise between the quantum efficiency (QE) and the photosensor response speed [11]. In the photo-sensing market, only silicon APS promises outstanding performance with high current amplification and high optical response [12,13,14,15,16]. This optical response can be extended to the visible spectrum.

Due to their inherently higher mobility than pure silicon (Si), silicon germanium (SiGe) thin films are better suited for use in photo-sensing and optical communication devices [17]. In particular, poly-SiGe (p-SiGe) films are a great candidate for Si–based photo-sensing due to their compatibility with Si and high optical absorption. Additionally, poly-SiGe films are also a promising alternative to amorphous-SiGe (a-SiGe) films due to their compatibility with Si, as well as their lower growth temperatures, higher mobility, and higher heat resistance coefficients. These are excellent properties for its production in the manufacturing industry [18,19,20,21,22]. The APS devices that are made with Poly-SiGe thin films are an efficient way to improve the detection performance in avalanche applications.

Due to the stress caused by the 4.2% SiGe lattice mismatch and significant temperature growth, deposition temperature is an important factor in the APS performance for poly-SiGe films [23]. Poly-SiGe films with a narrow bandgap lead to unacceptably high defect densities, especially through dislocations, in the deposited films, and most techniques in traditional solid-phase crystallization processes (such as molecular beam epitaxy, rapid thermochemical vapor deposition, or pulsed laser-assisted deposition techniques) have films with poor step coverage. S. Kobayashi et al. demonstrated that the flow ratio of working gas, H_2_ and Ar, affects the microstructure and optoelectronic properties of the formed poly-SiGe films [24]. The key role of hydrogen in the poly-SiGe layers growth process is to reduce the defects in the crystalline materials at a certain temperature [25]. On the growth surface, the high H-coverage reduces the number of Si dangling bonds. As a result, it can effectively increase the surface mobility of the grown precursor, allowing free radicals to find a more stable position, and promote crystallization. Additionally, the passivation effect of H_2_ can also be used for improving the grain boundary of the better efficiency in the device applications [26,27].

However, the state of crystallization depends on the Ge content or the deposition process, and the location of the nucleation site affects the microstructure of the films [28]. Controlled heterogeneous nucleation to form a poly-SiGe quantum confinement effect can be achieved by the thermal oxidation of the poly-SiGe film. Quantum dots allow precise size control and induce quantum confinement effects as they vary with electronic structure and band structure for optoelectronic applications [29,30]. The APS can improve the response rate due to its internal carrier multiplication mechanism. Since poly-SiGe films have a good ionization coefficient ratio, they are one of the best materials for the fabrication of an APS [31,32].

Numerous techniques for SiGe thin layer deposition for photo-sensing applications were investigated. These techniques included ultra-high vacuum and reduced pressure chemical-vapor deposition (UHVCVD or RPCVD) [33], low-pressure chemical-vapor deposition (LPCVD) [34,35,36], plasma-enhanced chemical-vapor deposition (PECVD) [37,38,39], and pulsed laser deposition (PLD) [40]. The key technology developed in this study is a selective LPCVD epitaxial growth of poly-SiGe thin films at low-temperature. A LPCVD can adjust the Ge content of the poly-SiGe film by changing the flow rate ratio of silane (SiH_4_) and germane (GeH_4_). The boundary layer deposited by a LPCVD reactor has a low molecular density [41], which is different from those of the other CVDs. This low-cost action forms conformal films with high-aspect step coverage, which is beneficial to the APS system with high detecting performance.

The poly-SiGe layers deposited on Si substrate we reference to other authors’ studies [42], the reason for the increase in grain size is not observed at annealing temperatures lower than 1000 °C. Colace et al. [43] and Luan et al. [44] effectively reduced thread dislocations by thermal annealing. According to the reference literature [42,43,44,45], the annealing temperature between 700 °C and 900 °C is adjusted for optimally transferring the surface morphology from the thermal energy to the crystalline structure. An as-deposited SiGe film has a poly-structure and will undergo post-annealing to improve its quality. Crystallization is induced at the interface between the layer and the crystalline substrate by thermal annealing in order to form a uniform layer.

In order to obtain the optimum APS growth conditions, we investigated various annealing parameters, such as the annealing temperature and annealing time of the poly-SiGe films. The proposed structure consists of indium-tin oxide (ITO)/poly-Si_0.8_Ge_0.2_ films and aluminum (Al)/silicon dioxide (SiO_2_)/Si-substrate, which is a reach-through (n^+^-n-p-p^+^) structure. The photocurrent of the APD element increases with an increase in illumination intensity. A structure of absorption layers that are combined with thin films of different annealing temperatures was developed for a larger output current ratio than traditional photo-sensors. By using a curve scanner (Tektronix 577 I-V) to measure the photocurrent of the device, we found that poly-Si_0.8_Ge_0.2_ structures have excellent current rates under illumination.

To obtain the optimal APS growth condition, we systematically studied different annealing parameters, such as annealing temperatures (700~900 °C) with hydrogen content for 30 min in poly-SiGe films. In this research, the optimal conditions for developing an APS structure, i.e., indium-tin oxide (ITO)/n^+^-n-p-p^+^ poly-Si_0.8_Ge_0.2_ films and aluminum (Al)/silicon dioxide (SiO_2_)/Si-substrate, was found and compared to that of a traditional APS. This investigation discusses not only the enhanced optical absorption, but also its other electrical properties. The developed APS has garnered significant interest as a sensing technology because of its potential to achieve high phonon responsivity and excellent electrical properties. Thus, the ultimate goal of this study is to integrate APS, which provides a route for low-cost, chip-scale, and photo-sensing systems.

## 2. Experiments and Measurements

Both oxidized, un-doped crystalline Si (100) wafers and conventional Si precoated with indium tin oxide (ITO) were used as substrates for the deposition of poly-SiGe films. The Si-substrate was selected for scanning electron microscope (SEM) analysis. A poly-SiGe thin film containing a n^+^-n-p-p^+^ alloy (total thickness 600 nm) was deposited by the LPCVD system.

Poly-Si_1-x_Ge_x_ films were coated with a RF power of 50 W, temperature of 620 °C, and total pressure of 40 Pa. The gas sources used consisted of SiH_4_, PH_3_, B_2_H_6_, and GeH_4_. The flow rate of SiH_4_ was set to 20 sccm, while the flow rate of GeH_4_ was varied from 300 sccm to 60 sccm. After the poly-SiGe film deposition, a 350 nm ITO layer was deposited using RF sputtering system. The ITO target used in the laboratory had the ratio of In:Sn = 9:1; argon (Ar) gas was used for sputtering.

Finally, the samples were treated with 60 sccm hydrogen content at anneal temperatures ranging from 700 °C to 900 °C for 30 min. The developed APS with an ITO (anode)/poly-Si_0.8_Ge_0.2_ films with a n^+^-n-p-p^+^ alloy and Al (cathode)/SiO_2_/Si-substrate structure was prepared using as-deposited and annealed poly-Si_0.8_Ge_0.2_ films. The thicknesses of each layer based on the growth-rate estimation were controlled, as described in our previous studies [46,47,48].

For experimental measurement, the chemical composition of the as-deposited films was quantitatively analyzed by an energy dispersive X-ray spectrometer (EDS, Noran Instruments, mod. Vantage v.1.2 made in Fitchburg, WI, USA). The process time was 5 s, and the spectrum range and corresponding lifetime of EDS parameters were operated at 0/10 keV and 100 s, respectively. The crystal structure of the poly-SiGe film was evaluated by a Rigaku X-ray diffraction (XRD, Siemens D5000 made in Germany) with the Ni-filtered Cu K_α_ radiation. The surface morphologies of the deposited films were examined by a SEM instrument (Hitachi S-4100 made in Japan). The working voltage, magnifying rate, and corresponding distance of SEM parameters were set at 15 KV, 30~220 K, and 9.6~16 mm, respectively. The film thickness was measured with an alpha-step-200 profile-meter supplied by Tencor. To measure the optical gain values of the proposed APS, current-voltage (I-V) curves were plotted with a curve tracer (Tektronix 577 made in USA). A band-pass filter (500–570 nm) was applied to detect the efficiency of the APS. The average transmission of a green filter used in the 536–550 nm range was 91.47%. The photo current (*I_p_*) of the developed devices was measured with a commercial tungsten light at a wavelength of 550 nm and 3 μW/cm^2^ power.

## 3. Results and Discussion

The crystallographic orientation of the film layers depends mainly on the temperature and thickness. In order to discuss the crystallinity of the thin films, the crystallographic orientation and the degree of strain relaxation of the deposited layers were determined by using an X-ray diffraction (XRD). Figure 1 displays the XRD patterns of the poly-Si_0.8_Ge_0.2_ films that were annealed from 700 °C to 900 °C for 30 min. The obtained signals correspond to the (111), (220), and (311) crystalline planes of the SiGe alloy, which are characteristics of a material with a SiGe polycrystalline. Due to a lack of thermal energy, only little crystallizations could be found without annealing.

Since the intensity and sharpness of the three diffraction peaks increased with an annealing temperature up to 800 °C, we also saw an increase in the crystallinity of the film. As the annealing temperature rose to 700 °C, the peak (111) started to increase, and the films began to crystallize. As the annealing temperature rose to 700 °C, an even higher increase in peak values of SiGe (111) and SiGe (220) were observed, while the peak value of SiGe (311) dropped by 25%. When the annealing temperature started to increase the crystallinity and the annealing temperature reached 800 °C, there was a tremendous increase in all three peak values of SiGe (111), SiGe (220), and SiGe (311) up to 100%, compared with those without annealing. This temperature was high enough to cause the thermal energy to trigger crystallization, especially for the SiGe (311) diffraction patterns. However, as the annealing temperature further increased up to 900 °C, all SiGe-intensity peaks reached the highest value for diffraction patterns: 167 for (111), 197 for (220), and 97 for (311), respectively. The electron diffraction patterns contained crystallographic orientation characteristic of the SiGe polycrystalline structure. We concluded that the SiGe film deposited at 800 °C is best for crystallographic orientation.

Since the poly-Si_0.8_Ge_0.2_ films employed in the APS were deposited directly in consequent, the surface morphologies of the deposited films may have affected the characteristics of the device. Figure 2a–d show the SEM images (2 × 2 μm^2^) of the poly-Si_0.8_Ge_0.2_ films with and without annealing at different temperatures. Based on the diagrams in Figure 2b,c, the grain size and surface roughness of the film morphologies initially increased with the increasing annealing temperature and reached saturation at 800 °C, then decreased dramatically as the annealing temperature rose to 900 °C, as seen in Figure 2d. This is attributed to the thermal energy provided by the annealing temperature that transferred the surface morphology and reached the most poly-crystalline with saturation at 800 °C [43,44,45]. However, at the annealing temperature of 900 °C, surface roughness decreased dramatically (Figure 2d). This is because the energy supplied at the high temperature generated defects for the deposition material. This phenomenon is consistent with the XRD patterns illustrated in Figure 1. Thus, with the LPCVD system, the best annealing temperature for the deposition of the best poly-crystalline film is 800 °C for 30 min.

The composition of the poly-Si_0.8_Ge_0.2_ layers was evaluated with an EDS chemical analysis and related to the contrast observed in the SEM image. Figure 3 depicts the X-ray counts from Ge and Si atoms, which represented the atomic percentage of each element. The atomic percentage of the Si:Ge ratio for the chemical composition was approximately 80:20 (denoted Si_0.8_Ge_0.2_). Meanwhile, the XRD spectra was applied to obtain the structure of the Si_0.8_Ge_0.2_ films, which showed both apparent Si and Ge peaks in accordance with the X-ray counts.

A schematic diagram of the APS structure designed for the present work is shown in Figure 4a. The proposed structure consisted of poly-Si_0.8_Ge_0.2_ films with n^+^-n-p-p^+^ layers inserted between the cathode Al and the anode ITO substrate. The APS in this study had a width of 200 μm, a length of 1000 μm, and a height of 600 μm. After the deposition process was completed, the contaminated regions were etched out by a buffer solution. Lastly, a 500 nm Al layer, made up of a cathode electrode, was evaporated and used as the mask of plasma etching to define the device area (~8 mm^2^). A cross-sectional SEM image of the sample annealed at 800 °C was taken (Figure 4b). In this image the poly-Si_0.8_Ge_0.2_ sandwiched structure exhibited clear and sharp boundaries between the layers with no threading dislocations, which was consistent with the proposed structure shown in Figure 4a. With the above results, we believe that the current fabrication process and degree of precision can be used in photo-sensing applications and optical communication systems.

The bottom-left axis of Figure 5 graphs the current-voltage (I-V) curves of the photo current (*I_p_*) under 3-μW/cm^2^ incident light among four APS devices treated with and without different annealing temperatures, i.e., 700 °C, 800 °C, and 900 °C for 30 min. It is worth noting that the *I_p_* of all the APS devices increased with increasing the reverse biased voltages (*V_R_*). All the APS photocurrents with the annealing treatment showed better values than those without the annealing treatment. The *I_p_* value annealed at 800 °C was the largest and increased by about 70%, compared with the none-annealing value.

In order to confirm that the photocurrent multiplication is effective for the APS device developed, the optical sensing gain ratio (*G*) was calculated using the following formula [49,50]:(1)$$G=\left[\frac{I_{p}-I_{d}}{q}\right]\times \left(\frac{E}{P_{in}}\right)$$
where *I_p_* is the photocurrent, *I_d_* is the dark-current, *P_in_* is the input power, *E* is the energy of the incident photon, and *q* is electron charge. The wavelength of the radiation can be used for the APS. Therefore, energy of the incident radiation is Planck’s constant *h* multiplied by the frequency (*λ*). The equation can be expressed as:(2)$$G=\left[\frac{I_{p}-I_{d}}{q}\right]\times \left(\frac{hc}{P_{in}\lambda }\right)$$
where *c* = 2.998 × 10^8^ m/s, *h* = 6.626 × 10^−34^ J·s, and *q* = 1.6 × 10^−19^ C. In Equation (2), the optical gain ratio *G* is calculated from the measured data *P_in_* and *λ*. The top-right axis of Figure 5 shows the gain ratio versus the intensities of the incident light. Both the *P_in_* and *λ* were calculated for the investigated APS with the peak wavelength at 550 nm, under the maximum 27-V reverse biasing voltage. In the top-right axis of Figure 5, the biased voltage reached saturation at 3 μW/cm^2^, while the APS at 800 °C annealing temperature processed the maximum optical gain ratio of 143 approximately 700% higher than the none-annealing type. These results agree with the experimental data in the bottom-left axis of Figure 5.

The responsivity (***R****_resp_*) is defined as the ratio of the detected photocurrent (*I_p_*) and the absorbed optical power (*P_opt_*). The top-right axis of Figure 6 describes the ***R****_resp_*-values (in A/W) as a function of the biasing voltages with respect to the different annealing conditions of the developed APS devices at 3-μW/cm^2^ incident power and 550 nm wavelength. Moreover, the responsivity at the 800 °C annealing condition of the developed poly-Si_0.8_Ge_0.2_ APS device also increased to about 96% as compared with the none-annealing APS, thus confirming its good photo-sensing characteristics applied to the atmospheric particulate matter detection.

In order to represent the photoelectric effect of the developed APS devices, we defined the quantum efficiency (*η_QE_*, in %), which is one of the key factors and expressed by the following Equation (3), as the generated electron-hole pairs for each incident photon energy (*h**ν*) [51,52]:(3)$$\eta_{QE}=\frac{r_{e}}{r_{p}}=\left(\frac{P_{m}}{q}\right)={\left(\frac{P_{opt}}{hv}\right)}^{-1}$$
where *r_p_* and *r_e_* are the photons per second of the incident and the generating photons, respectively. *P_m_* is the photocurrent before multiplication. The bottom-left axis of Figure 6 shows the *η_QE_*-*V_R_* curve of the developed APS devices at 3-μW/cm^2^ incident power and 550 nm wavelength. The highest response peaked at 800 °C annealing condition and improved greatly (96%) as compared to the none-annealing APS. It is attributed that all the primary generated electrons are assumed to experience a lumped multiplication process immediately in the generation of the secondary electron-hole pairs [52,53].

The schematic diagram for the photo-sensing and optical communication applications is shown in the inset of Figure 7. The proposed APS was connected to an IC (LM358), which was used as a buffer to prevent attenuation. The resistor-capacitor (***R***_f_-***C***_f_) cut-off filter was connected before and after the buffer to provide actual DC input. The most important figure of merit in the APS characteristics is the avalanche multiplication factor (*M*) [54], which is similar to those of an APS reverse-biased voltage (*V_R_*) with a ballast resistance (*R*) and a breakdown voltage (*V_B_*). This can be expressed with the equation below [55]:(4)$$M=\frac{1}{1-{\left[\frac{\left(V_{R}-RI_{P}\right)}{V_{B}}\right]}^{n}}$$
where the exponent *n* is related to the impurity distribution in the semiconductor material and the wavelength of the incident light (*hν*). Figure 7 plots the *M*-values calculated from Equation (4), where the M-values are related to the voltage ratio (*V_APS_*/*V_B_*) from different *n*-values (=3~6). Normally, the junction voltage (*V_APS_* = *V_R_* − *RI_p_*) is smaller than the biased voltage under the developed APS operation. The *M*-values of the plot slowly increased with the small *V_APS_*/*V_B_*-ratios, and rapidly rose with the *V_APS_*/*V_B_*-value approaching 1. This mechanism can be explained as follows: since the total quantity of electron-hole pairs in the light absorption region is finite, the number of photo-generated carriers will be saturated gradually when the biasing voltage reaches a certain value.

The poly-Si_0.8_Ge_0.2_ layers exhibit excellent electrical and optical properties for device applications due to proper annealing. This study found that the enhancement of the device performance can be related to the annealing parameters in the Si_1−x_Ge_x_ film (SEM comparison shown in Figure 3), indicating that the illumination significantly increases the current generated by the phonon carriers within the Si-base structure. For the developed APS applications in Figure 7, the M-parameter curves by the n-values (from 3 to 6) of Equation (4) were plotted with increasing voltage ratios (*V_APS_*/*V_B_*). Furthermore, the optimal avalanche photocurrent (M value) applied on the inset circuit diagram can be obtained by taking *n* = 3, which is agreed on the *R*-value extracted from the slope of the I-V curve at the optimal 800 °C annealing condition in the bottom-left axis of Figure 5. Furthermore, the experimental results under *n* = 3 conditions are included for the purpose of comparison. A quite good fitness can be found between the calculated and measured results in Figure 7.

## 4. Conclusions

This paper discusses the design and fabrication of a high-performance APS with a n^+^-n-p-p^+^ alloy/SiO_2_/Si-substrate structure using a LPCVD system and the optimal annealing conditions, which are 800 °C for 30 min, to deposit good poly-Si_0.8_Ge_0.2_films. Under 3-μW/cm^2^ incident light (with peak wavelength at 550 nm) and 27-V biased voltage, the developed APS possesses a maximum photocurrent, responsivity, and quantum efficiency of almost 70%, 96%, and 85%, when compared to those of the none-annealing APS type. These excellent detection performances indicate that the proposed APS is a candidate for a low-cost chip for photo-sensing and optical communication applications.

## Figures and Tables

**Figure 1 sensors-22-01243-f001:**
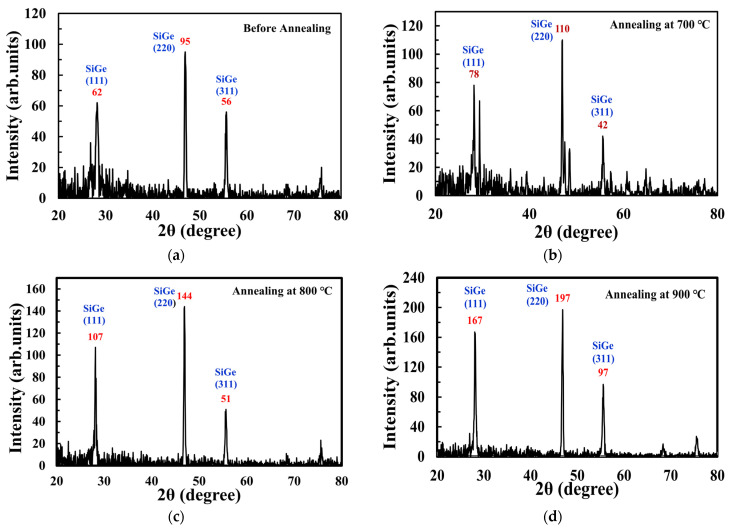
X-ray diffraction pattern of a polySi_0.8_Ge_0.2_ film prepared (**a**) without annealing, and at different annealing temperatures of (**b**) 700 °C, (**c**) 800 °C, and (**d**) 900 °C for 30 min.

**Figure 2 sensors-22-01243-f002:**
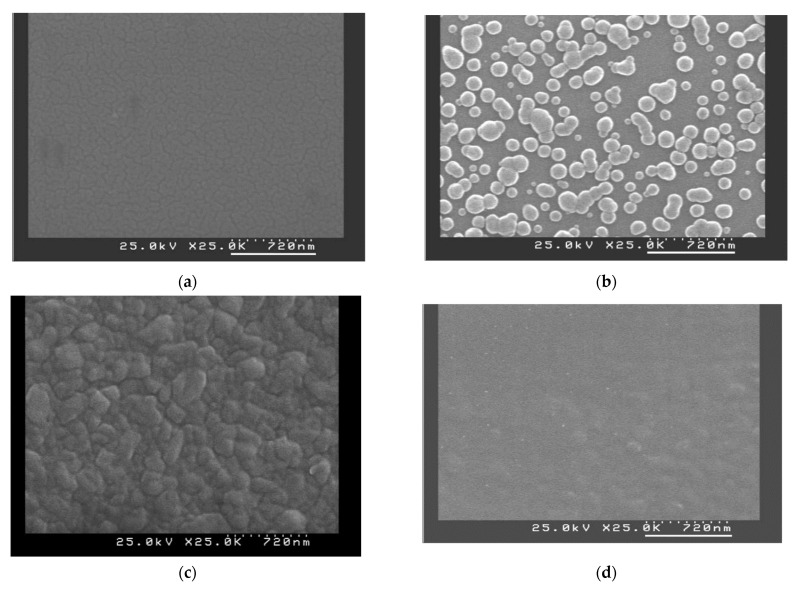
SEM images of poly-Si_0.8_Ge_0.2_ film deposited (**a**) without annealing, and at different annealing temperatures of (**b**) 700 °C, (**c**) 800 °C, and (**d**) 900 °C for 30 min.

**Figure 3 sensors-22-01243-f003:**
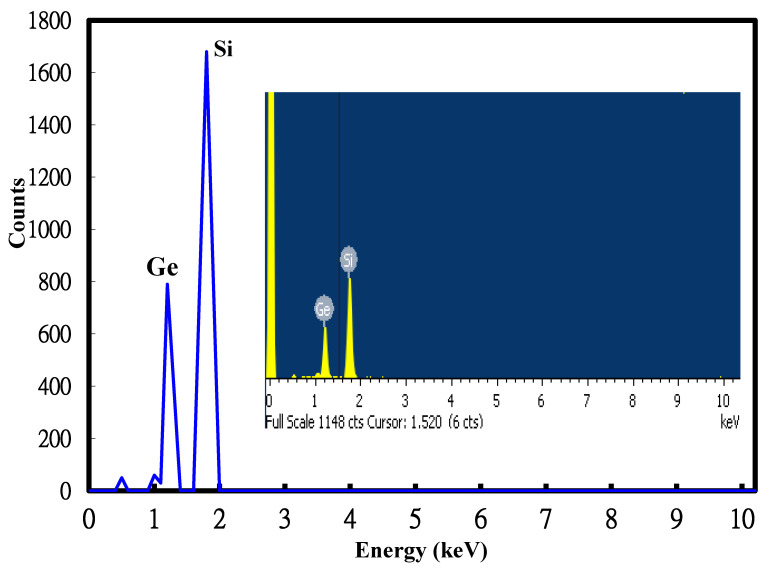
Energy dispersive X-ray spectrometer (EDS) analysis of the deposited Si_0.8_Ge_0.2_ films.

**Figure 4 sensors-22-01243-f004:**
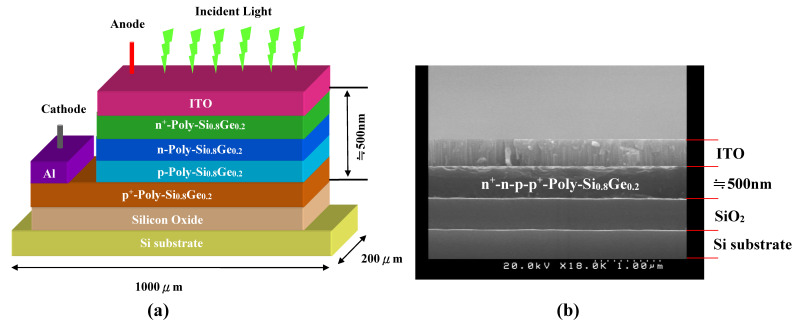
(**a**) Indium tin oxide (ITO)/n^+^-n-p-p^+^ polySi_0.8_Ge_0.2_ film and schematic APS device with aluminum (Al)/silicon dioxide (SiO_2_)/Si-substrate structure and (**b**) cross-section SEM photo of the developed APS (30 mm^2^ dimension).

**Figure 5 sensors-22-01243-f005:**
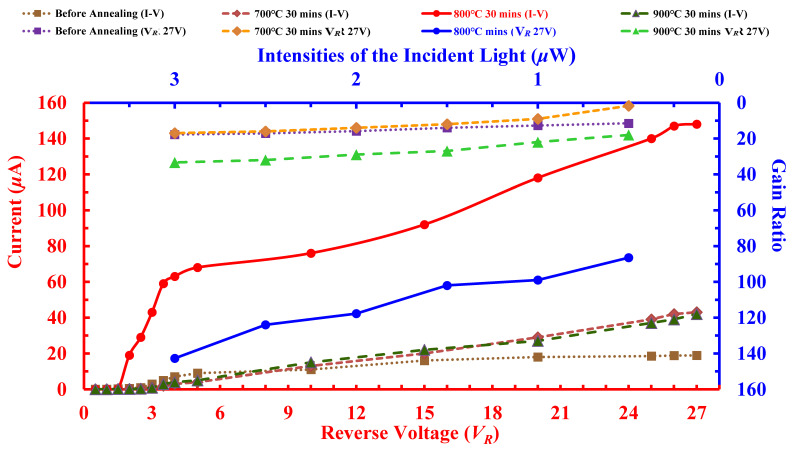
The current-bias voltage (I-V) curves of poly-Si_0.8_Ge_0.2_ films under 3-μW incident light power (bottom-left axes) and gain ratio curves versus using a reverse bias voltage up to 27 V (top-right axes). The APS devices were prepared with as-deposited and annealed poly-Si_0.8_Ge_0.2_ films at 700 °C to 900 °C for 30 min.

**Figure 6 sensors-22-01243-f006:**
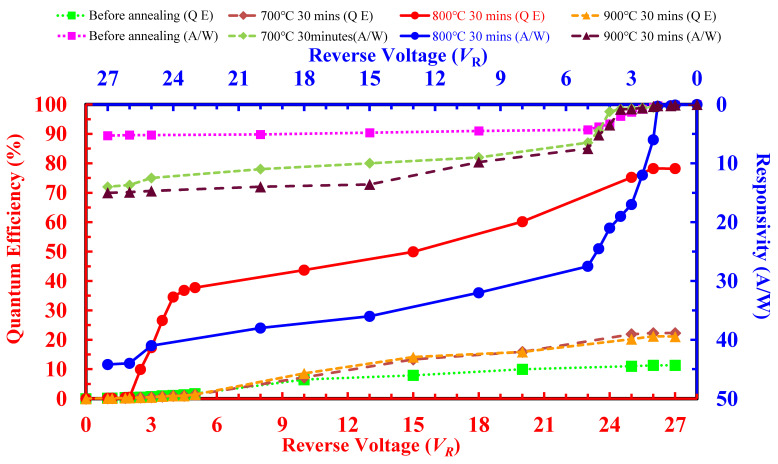
The variation of the responsivity (***R****_resp_*, top-right axes) and quantum efficiency (*η_QE_*, bottom-left axes) with the 3-μW/cm^2^ incident light at reverse bias up to 27 V. The APS devices were prepared with as-deposited and annealed poly-Si_0.8_Ge_0.2_ films at 700 °C, 800 °C, and 900 °C for 30 min.

**Figure 7 sensors-22-01243-f007:**
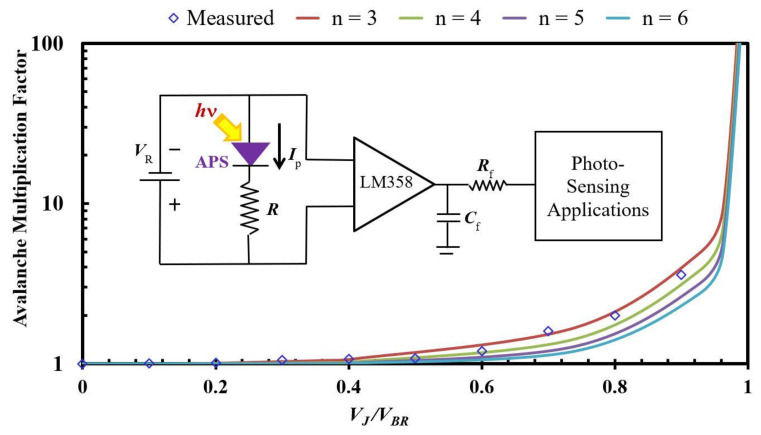
Relationship between the avalanche multiplication factor (*M*) and the voltage ratio (*V**_APS_*/*V**_B_*) from different *n*-values (=3~6), which was measured from the circuit diagram for the photo-sensing and optical communication applications, as shown in the inset. These theoretical calculations are also compared with the experimental results under *n* = 3 condition.

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
