# Peer review of "Performance of High Efficiency Avalanche Poly-SiGe Devices for Photo-Sensing Applications"

_sensors, 2022, doi:10.3390/s22031243_

Round 1
Reviewer 1 Report
Please see my comments in the attachment.
Best Regards

Author Response
Dear Chief Editor & Dr. Li:
Enclosed please find one files of the paper (R3AllAPS-1545069.doc) entitled above, by Y.-T. Cheng et.al. In the revised one, all the comments of the reviewers have been overcome and marked with red/bold words. Hope this revised one can be accepted to publish on the Sensors Journal.
In addition, the special revisions per reviewer’s comment (AnswerQueries-1545069.pdf) have been attached for reviewers’ convenience to check.
Your kind assistances in dealing with this matter are my most appreciated.
Happy New Year Season --
Best regards from Sincerely Yours,
Jyh-Jier HO, Ph.D. 01/13/2022

Reviewer 2 Report
The authors have reproted the new strucuture of APS device, and exhibited good behaviors. I think this manuscript can be accpeted after the minor revised. There are some issues as following.
- Figure 5,why the plot of 900℃ 30 mins (I-V) is below 0 μA? Please explain.
- Figure 6,why the plot of 900℃ 30 mins (QE) is below 0 %? Please explain.
- I wonder why choose the light wavelength of 550 nm?
- The power density of 550 nm is 3μW?May be it is 3μW/cm2 or 3μW/m2?
- Why the n+-n-p-p+ alloy/SiO2/Si-substrate exhibit excellent photocurrent, responsivity and quantum efficiency?
Author Response
Dear Chief Editor & Reviewer 2:
Enclosed please find one files of the paper (R3AllAPS-1545069.doc) entitled above, by Y.-T. Cheng et.al. In the revised one, all the comments of the reviewers have been overcome and marked with red/bold words. Hope this revised one can be accepted to publish on the Sensors Journal.
In addition, the special revisions per reviewer’s comment (AnswerQueries-1545069.pdf) have been attached for reviewers’ convenience to check.
Your kind assistance in dealing with this matter are my most appreciated.
Happy New Year Season --
Best regards from Sincerely Yours,
Jyh-Jier HO, Ph.D. 01/13/2022

Reviewer 3 Report
Comments from Reviewer: Sensors-1545069
- Comments to Authors:
Title:
Performance of High Efficiency on Avalanche Poly-SiGe Devices for
Photo-sensing Applications
Authors:
Yuang-Tung Cheng, Tsung-Lin Lu, Shang-Husuan Wang, Jyh-Jier Ho,
Chung-Cheng Chang, Wen-Hao Huang, Jiashow Ho
General comments:
[Summary of this manuscript]
Avalanche diode device is an extremely important for various optical semiconductor device manufacturing. This manuscript presents unique study on Poly-SiGe film deposition on silicon substrate using by chemical vapor deposition with and without additional annealing.
Based on this result, the authors concluded that optical property such as quantum efficiency of avalanche Poly-SiGe diode device strongly depends on additional annealing condition such as temperature.
As a result, additional annealing treatment is enhancement of quantum efficiency of avalanche Poly-SiGe film diode device. Therefore, it is very important to deep understanding of additional annealing behavior of avalanche Poly-SiGe film diode device for high quantum efficiency optical device engineering and materials science point of view.
[Evaluation of this manuscript]
This manuscript reported that the annealing behavior of avalanche Poly-SiGe film deposited on silicon substrate fabricated by chemical vapor deposition with and without additional annealing.
This study is extremely challenging and informative fundamental experimental data for optical device engineering application.
However, this manuscript has some problems about explanation of experimental data and results. Therefore, the reviewer strongly recommends the authors to reconsider the explanation of experimental data and results. If the authors can clearly answer the specific comments, the reviewer will reconsider this manuscript.
Specific comments:
- What is the material science finding of this manuscript?
It is very difficult for reviewer to understand this manuscript of “optical device application and material scientific findings” for avalanche Poly-SiGe semiconductor device fabrication.
Reviewer strongly recommends improvement of introduction section and additional information.
- Reviewer strongly recommends that the more detail and clear explanation of poly-Si1-xGex film deposition temperature using low pressure chemical vapor deposition before additional annealing.
Why do the author choice the additional annealing condition such as 700C, 800C and 900C.
- Reviewer strongly recommends that the more detail and clear explanation of additional annealing in hydrogen atmosphere for SiGe deposition film crystal quality. What is “hydrogen annealing effect” of SiGe deposition film crystal quality?
- Do the author have SEM observation result (measurement data) of SiGe deposition film surface crystal quality depends on additional annealing duration time such as 800C X 1hour. It is very important to SEM observation result of the deposition film surface quality for optimize deposition film conditions.
- Why do the poly-Si1-xGex film crystal quality after 900℃ annealing compare than that of after 800℃ annealing (See page number 3: How-ever, as the annealing temperature further increased up to 900℃, show all three peak val-ues have drastically dropped compared with 800 ℃. For SiGe (311), it is even lower than that without annealing. This means up to certain level the thermal energy might bring damage to the crystallization)
Reviewer strongly recommends that the more detail and clear explanation of poly-Si1-xGex film crystal quality after 900℃ additional annealing.
Author Response
Dear Chief Editor & Reviewer 3:
Enclosed please find one files of the paper (R3AllAPS-1545069.doc) entitled above, by Y.-T. Cheng et.al. In the revised one, all the comments of the reviewers have been overcome and marked with red/bold words. Hope this revised one can be accepted to publish on the Sensors Journal.
In addition, the special revisions per reviewer’s comment (AnswerQueries-1545069.pdf) have been attached for reviewers’ convenience to check.
Your kind assistances in dealing with this matter are my most appreciated.
Happy New Year Season --
Best regards from Sincerely Yours,
Jyh-Jier HO, Ph.D. 01/13/2022

Round 2
Reviewer 1 Report
I very much thank the authors for the revision. The authors' aim in this work is to propose a poly SiGe APD structure to be used in the next-generation low-cost, large scale photo-sensing applications. However, Ge-on-Si and SiGe APDs have already been developed quite alright in the last two decades, and the advantage of Poly SiGe material is not very clear to me when I still read the text. A comparison figure/table of this device with Ge and SiGe APDs is definitely required.
At the moment, their device is quite large compared to the state-of-the-art. Information about its scalability to um level, and how the defect density will vary at small devices are unknown. Besides, dark current levels are not better than Ge or SiGe APDs. Gain-bandwidth product info is missing, which is a crucial parameter to present together with gain/responsivity data. Therefore, these issues should be addressed/measured to see if poly SiGe is an alternative material system to replace Ge-on-Si or SiGe APDs. It would be also better to make a structure with separate absorber and multiplication method to make fair comparison with the literature. In conclusion, this work is important, but, in my opinion, it is at the very early stage. In the future, I believe that the authors will improve this work much better.
One final note: Regarding Fig.7, I think this figure is a bit misleading. When I read the text and your answers, I understand that M parameter is purely calculated. However, there is a setup in the figure, which gives the impression that you measured the M parameter. What has been done with this setup is still not very clear to me.
Author Response
Dear Chief Editor & Reviewer 1,
Enclosed please find one files of the paper (Round2Sensors-1545069.docx) entitled above, by Y.-T. Cheng et.al. In the revised one, all the comments of the reviewers have been overcome and marked with purple/bold words. Hope this revised one can be accepted to publish on the Sensors Journal.
In addition, the special revisions per reviewer’s comment (Round2-AnswerQueries-1545069.pdf) have been attached for reviewers’ convenience to check.
Your kind assistance in dealing with this matter is my most appreciated.
Happy New Year Season --
Best regards from Sincerely Yours,
Jyh-Jier HO, Ph.D.

Reviewer 3 Report
2022.01.19
Comments from Reviewer: Sensors-1545069
- Comments to Authors:
Title:
Performance of High Efficiency on Avalanche Poly-SiGe Devices for
Photo-sensing Applications
Authors:
Yuang-Tung Cheng, Tsung-Lin Lu, Shang-Husuan Wang, Jyh-Jier Ho,
Chung-Cheng Chang, Wen-Hao Huang, Jiashow Ho
General comments:
[Summary of this manuscript]
I read above the revised manuscript from authors carefully.
I understand that the authors can clearly answer the specific comments from reviewer.
Therefore, my opinion is accepted this revised manuscript by Sensors MDPI.
Author Response
Dear Chief Editor & Reviewer 3,
Enclosed please find one files of the paper (Round2Sensors-1545069.docx) entitled above, by Y.-T. Cheng et.al. In the revised one, all the comments of the reviewers have been overcome and marked with purple/bold words. Hope this revised one can be accepted to publish on the Sensors Journal.
In addition, the special revisions per reviewer’s comment (Round2-AnswerQueries-1545069.pdf) have been attached for reviewers’ convenience to check.
Your kind assistance in dealing with this matter is my most appreciated.
Happy New Year Season --
Best regards from Sincerely Yours,
Jyh-Jier HO, Ph.D.
=================
Revision per Reviewer’s Instructions
(MS#: sensors- 1545069)
Reviewer 3’s comment:
I understand that the authors can clearly answer the specific comments from reviewer. Therefore, my opinion is accepted this revised manuscript by Sensors MDPI.
Ans: We are appreciated deeply to the reviewer for preproduction review on the round 2 report.
===================
Finally, we are appreciated deeply to the reviewers and the Sensors Staff for preproduction review and technical assistance in the original text. Following these suggestions, we have corrected them with bold/purple-marked words as following in the revised text. Meanwhile, the paper writing has been corrected and checked them carefully in the revised text.